# Influence of Sociodemographic Variables and Healthy Habits on the Values of Insulin Resistance Indicators in 386,924 Spanish Workers

**DOI:** 10.3390/nu15245122

**Published:** 2023-12-16

**Authors:** Miguel Mestre Font, Carla Busquets-Cortés, José Ignacio Ramírez-Manent, Pilar Tomás-Gil, Hernán Paublini, Ángel Arturo López-González

**Affiliations:** 1ADEMA-Health Group, Instituto Universitario en Ciencias de la Salud, University of Balearic Islands, 07122 Palma, Spain; m.mestre@eua.edu.es (M.M.F.); c.busquets@eua.edu.es (C.B.-C.); p.tomas@eua.edu.es (P.T.-G.); h.paublini@eue.edu.es (H.P.); angarturo@gmail.com (Á.A.L.-G.); 2Familiy Medicine, Balearic Islands Health Service, 07003 Palma, Spain

**Keywords:** insulin resistance, smoking, physical exercise, social class, Mediterranean diet

## Abstract

Background: Insulin resistance (IR) is an alteration of the action of insulin in cells, which do not respond adequately to this action, leading to an increase in blood glucose levels. IR produces a very diverse clinical picture and increases the cardiometabolic risk of the population that suffers from it. Among the factors that influence IR are genetics, unhealthy lifestyle habits, overweight, and obesity. The objective of this work was to determine how different sociodemographic variables and healthy habits influence the values of different scales that assess the risk of presenting IR in a group of Spanish workers. Methods: An observational, cross-sectional, descriptive study was carried out in 386,924 workers from different Spanish regions. Different sociodemographic variables and lifestyle habits were studied (age, social class, educational level, smoking, Mediterranean diet, physical exercise) along with their association with four scales to evaluate the risk of insulin resistance (TyG index, TyG-BMI, METS-IR, TG/HDL-c). To analyse the quantitative variables, Student’s *t* test was used, while the Chi-squared test was used for the qualitative variables. A multinomial logistic regression analysis was performed, calculating the odds ratio with its 95% confidence intervals. The accepted level of statistical significance was set at *p* < 0.05. Results: In the multivariate analysis, all variables, except educational level, increased the risk of presenting high values on the IR risk scales, especially a sedentary lifestyle and low adherence to the Mediterranean diet. Conclusions: Our results demonstrate an association between the practice of regular physical exercise and a reduction in the risk of IR; a strong role of the Mediterranean diet as a protective factor for IR; an association between aging and increased IR, which has also been suggested in other studies; and, finally, a relationship between a low socioeconomic level and an increase in IR.

## 1. Introduction

Insulin resistance (IR) is an alteration of the action of insulin in cells, which do not respond adequately to this action, leading to an increase in blood glucose levels [1,2]. This pathology is worrying in our society, since it is a problem that is increasing in countries with a higher standard of living, which is often associated with obesity [3,4]. In response to this situation, not only do the pancreatic islets continue producing insulin, but they also increase its synthesis, thus elevating plasma levels, which is a sign of this dysfunction. This increase in circulating insulin raises insulin resistance at the tissue level, thereby increasing hepatic glucose production and decreasing muscle utilization, producing a vicious cycle [5,6]. When this situation lasts over time, different organs are affected, causing multiple disorders in the body [7].

IR can present a very diverse clinical picture that generally includes fatigue [7,8]; increased appetite [7]; difficulty in losing weight [7]; hypertension [9]; dyslipidaemia [10]; cardiovascular diseases [11,12]; dark dermal spots, especially on the neck and armpits [13,14]; lately, it has been associated with psoriasis, which it may precede in time [15]; different mental illnesses such as schizophrenia [16], bipolar disorder [17] or major depressive disorder [18,19]; cognitive impairment [20]; and various clinical entities such as polycystic ovary [21,22] and sleep apnoea [23,24]. Sometimes, IR can be asymptomatic until the very late stages of the disease [25].

Among the factors involved in the genesis of IR, it is worth noting genetics [26]; unhealthy lifestyle habits such as little physical activity [27], excessive and high-calorie feeding [28]; fatty liver and overweight [29]; and obesity, which is considered as having a body mass index (BMI) ≥ 30 kg/m^2^, which constitutes the most important risk factor for developing type 2 diabetes (DM2) and IR [30,31].

The correct approach to IR should include variations in one’s lifestyle [32,33] that incorporate physical activity [34,35,36], healthy eating, and weight loss if necessary [37,38]. In certain situations, it may be necessary to implement pharmacological treatment [39,40] to correctly control glycaemic levels. It is essential to achieve adequate control of IR because, otherwise, serious complications could appear [41,42], such as type 2 diabetes, heart disease [43,44], cerebrovascular accidents [45], and a state of chronic inflammation and prothrombotic activity [46,47].

The objective of this work was to determine how different sociodemographic variables and healthy habits influence the values of different scales that assess the onset risk of IR in a group of Spanish workers.

## 2. Methods

Our research is based on an observational, transversal, descriptive study carried out comprising 386,924 workers, from practically all labour sectors, who carried out their work activity in different Spanish regions: 232,814 men and 154,110 women. The workers selected for this study consisted of people who attended the periodic medical examinations carried out annually in the different participating companies. Selection was carried out between January 2019 and June 2020.

### 2.1. Inclusion Criteria

Being between 18 and 69 years old.Having an employment contract with one of the companies participating in this study.Agreeing to participate in this study.Allowing the use of the data for epidemiological purposes.

The data from the workers’ flowchart, once the inclusion criteria were applied, are shown in Figure 1.

### 2.2. Determination of Variables

The people at the occupational health units of the participating companies were responsible for obtaining the necessary data to carry out this study. The data were collected through the following:-Anamnesis. Owing to an exhaustive clinical history, the data on sociodemographic variables (age, sex, social class, and level of education) and healthy habits (tobacco, alcohol, Mediterranean diet, and physical activity) were collected.-Anthropometric and clinical determinations. These included height, weight, waist circumference, and systolic and diastolic blood pressure.-Analytical determinations. Lipid profiles and glycaemia were determined.

In an attempt to avoid possible biases in this study, the standardization of the measurement techniques and the collection of the variables was carried out.

Height and weight were obtained with participants in a standing position and in underwear, arms hanging, and chest and head aligned. A SECA model scale/height meter was used. Data were expressed in centimetres and kilograms.

To assess abdominal waist circumference, a SECA model measuring tape was used, placed at the level of the last floating rib and parallel to the floor. The participant stood with the abdomen as relaxed as possible. Hip circumference was measured in the same position by placing the measuring tape parallel to the floor at the level of the widest part of the gluteal area.

Blood pressure figures were determined using an OMROM-M3 model blood pressure monitor. For a correct determination, the person rested in a sitting position for at least 10 min beforehand. The cuff was placed around the arm so that it fitted well without tightening too much. For this purpose, cuffs of different widths and sizes were available. Three consecutive determinations were made with a separation of one minute between them. The figure considered is the average of the three determinations.

Data on blood parameters were obtained via venepuncture after a prior 12 h fast. Samples were processed and kept refrigerated for proper conservation for a period of no more than 48–72 h. Analyses were carried out in reference laboratories that used the same methodology. Enzymatic techniques were used to determine triglycerides, total cholesterol, and blood glucose, while precipitation techniques were used to determine HDL cholesterol. LDL cholesterol values were estimated indirectly by applying the Friedewald formula if triglyceride values did not exceed 400 mg/dL. If this value was exceeded, LDL was determined directly. All analytical variables were expressed in mg/dL.

The risk level of insulin resistance was established using a series of scales, as follows:Different scales were calculated to evaluate the risk of insulin resistance (IR).Metabolic insulin resistance score (METS-IR) [48].

METS-IR = Ln [(2 glycaemia) + triglycerides] BMI)/(Ln[HDL-c]). Values were considered high from 50 up.TyG index [49] and its variants:○TyG index = Ln [triglycerides (mg/dL) glycaemia (mg/dL)/2]. Values were considered high from 8.72 up in men and 8.67 up in women [50].○TyG-BMI was obtained by multiplying the TyG index by the BMI. Its cut-off point was 191.53 [51].○TyG-waist circumference [52]. This was obtained by multiplying the TyG index by the waist circumference.

Triglycerides/HDL-c [53]. Values were considered high from 2.4 up. This was obtained by dividing the value of triglycerides by the value of HDL cholesterol.

Sex was established as a dichotomous variable: man and woman.

Age was calculated by subtracting the date of medical examination from the date of birth.

The educational level considered was the highest of those that had been fully completed. Three levels were established: primary studies, secondary studies, and university studies.

Social class was determined by applying the criteria of the Spanish Society of Epidemiology, based on the type of work included in the national classification of occupations for 2011 (CNO-11) [54]. Three levels were established:-Social class I. This includes management personnel, professionals with university training, professional athletes, and artists.-Social class II. This includes intermediate professions and qualified self-employed workers.-Social class III. This includes low-skilled workers.

In our study, we considered a person a smoker if they had consumed any form of tobacco at least once every day in the previous month or had stopped smoking less than a year before.

Adherence to the Mediterranean diet was established using the questionnaire applied in the PREDIMED study [55], which consists of 14 questions rated 0 or 1 points. Values from 9 points indicate high adherence [56].

Level of physical activity was determined using the International Physical Activity Questionnaire (IPAQ) [57], a self-administered questionnaire that assesses physical activity carried out in the previous week.

### 2.3. Statistical Analysis

To analyse the quantitative variables, Student’s *t* test was used, determining the means and standard deviations. When the variables were qualitative, the Chi-squared test was used, calculating the prevalence. A multinomial logistic regression analysis was performed, calculating the odds ratio (OR) with its 95% confidence intervals. The statistical analysis was performed with the SPSS 28.0 program. The accepted level of statistical significance was set at *p* < 0.05.

## 3. Results

Table 1 presents the anthropometric, clinical, analytical, sociodemographic, and healthy habit data of the 386,924 workers in this study. The average age of the participants was slightly over 39 years. The variables show more unfavourable values in the group of men, except for LDL cholesterol. The study sample comprised a total of 60.2% men and 39.8% women. The majority group was between 30 and 49 years of age, while most people belonged to social class III and completed primary education. Of the total, 45.5% of men and 52.2% of women performed regular physical activity; 51.4% of women had a high adherence to the Mediterranean diet, whereas this percentage dropped to 41% in men; meanwhile, 37.1% of men and 33% of women were smokers.

Table 2 shows the mean values of the different insulin resistance risk scales included in this study. In the four scales, and in both sexes, the values increased with age. A rise in these values was also seen as the social class or educational level descended. Likewise, it can be observed that people with a low adherence to the Mediterranean diet, sedentary people, and smokers also had higher values on the IR risk scales. In all scales, the average values were higher in men. The differences observed in all cases were statistically significant.

Table 3, which shows the prevalence of high values of all IR scales, reveals the same model as the one discussed for the average values; that is, a greater prevalence of high-risk IR values is present as age increases, as socioeconomic and educational level descend, in smokers, and in people with low adherence to the Mediterranean diet and a low level of physical activity. As mentioned above, values were higher in men and all the observed differences were statistically significant.

Table 4 presents the results of the multivariate analysis using multinomial logistic regression. All sociodemographic variables—except educational level—influenced the appearance of high values on the IR risk scales. Tobacco consumption, a sedentary lifestyle, and low adherence to the Mediterranean diet also increased the risk of IR with the four scales. The variables that had the most influence—that is, those with the highest odds ratios—were sedentary lifestyle and diet.

## 4. Discussion

In our sample, we studied 386,924 participants of both sexes, all of whom were from Spain (60% men and 40% women). All of them corresponded to workers between 18 and 69 years of age, from the largest companies in Spain, predominantly hospitality, construction, commerce, health, public administration, transport, education, industry, and cleaning. For this study, they were stratified into age groups by decade, educational level, and social class. Most of the participants in our study were between 30 and 49 years old, belonged to social class III, and completed primary education.

Although we are aware that the most accurate way to assess insulin resistance is the HOMA-IR (insulin resistance index) [58], the enormous size of the sample makes the cost of said measurement unaffordable. However, the three indices used have been validated and we consider them to be useful in assessing the risk of insulin resistance.

Additionally, several studies have established a positive correlation between waist circumference and insulin resistance [59,60]. In our study, one of the three formulas used, the TyG index, takes this variable into account.

In the multinomial logistic regression analysis, all the formulas used to assess the risk of insulin resistance were found to present a higher risk in men than in women, which agrees with the published bibliography [61,62,63].

In our sample, 37% of men and 33% of women were smokers. These data are somewhat higher than those obtained in the European Health Survey in Spain in 2020, where the smoking population was concentrated between 25 and 64 years of age with percentages around 30% in men and 20% in women [64]. Now, if we take into account the fact that the majority of our sample belongs to a lower social class (77.3% men and 59.8% women), this could justify a higher prevalence of active smokers, since several studies have found that smoking is more common in people belonging to lower socioeconomic levels, where smoking is more accepted within their social and work networks, with less feeling of guilt and less social support. As is the case of the OEDA labour survey (2020) of the Spanish Ministry of Health, in which 35.6% of workers smoke daily, the typical smoker is a man under 24 years of age, with a low educational level, and a manual worker in hospitality, construction, sales, or unemployed [65]; moreover, social classes with a higher socioeconomic level and a higher level of education are more likely to give up smoking if they have started it, whether due to feelings of guilt, social norms, and/or family pressures [66,67,68,69]. In our study, we found a relationship between the four IR indices and smoking, with an increased risk in smokers—which has also been described in other studies [70].

Regarding physical exercise, 45% of men and 52% of women in our study carried out physical activity on a regular basis according to the results obtained with the IPAQ questionnaire. These results are also higher than those obtained in the European Health Survey in Spain in 2020, where the national result obtained was 26.5% of people over 15 years of age. However, the survey results in some autonomous communities exceed 40% of people who carry out regular physical activity in their free time (Navarra, Basque Country, Santander), which is closer to the results of our study [71]. Even so, we must bear in mind the fact that the population in our sample is over 18 years old, and that in the case of the European Health Survey in Spain in 2020, the responses are more subjective and a person may consider that they do not do enough physical activity, by not taking into account certain activities of everyday life. In the IPAQ, however, even though it is a self-administered questionnaire, the person can include certain activities, such as walking or carrying weights, which are assessed by the questionnaire. Consequently, it can provide a higher level of physical activity. Lack of regular physical activity has a very high OR in the four formulas used, especially with high TyG-BMI index, which implies a strong association between IR and the lack of physical activity. A meta-analysis carried out in 2019 found an association between performing physical exercise and reducing IR. However, it also stated that all the studies included in the meta-analysis had a low sample size and therefore encouraged studies with a large sample size [27]. In our study, the sample has a high population of both sexes, and a very high OR in the four formulas used, ranging between 21.10 and 78.77, with narrow 95% confidence intervals.

Another lifestyle factor that can influence insulin resistance is diet [72,73]. There is, in fact, no single “ideal” diet. Indeed, many diets are capable of satisfying the nutritional needs of our body. A person’s balanced diet is made up of the amount of food that he or she must eat each day to achieve an optimal nutritional state, that is, a supply of nutrients that ensure good individual health [74,75,76]. However, an unbalanced diet with an abuse of processed carbohydrates and saturated fats enables the development of obesity and IR [77,78]. In our case, we evaluated it in relation to participants’ adherence to the Mediterranean diet, whose health benefits include a reduction in the risk of cardiovascular diseases, a reduction in overweight, better glycaemic control, prevention of cognitive deterioration, an anti-inflammatory effect, and protection against some types of cancer [79,80,81]. The results obtained in the multivariate analysis show a strong association, as a protective factor, between the Mediterranean diet and the four IR assessment scales.

Age is another factor that is associated with both metabolic syndrome and insulin resistance [82], the latter forming one of the components of the metabolic syndrome. During aging, a series of metabolic changes occur that increase over the years and produce changes in the function and structure of tissues and their cells. In 1988, Harman [83] established the theory of free radicals, according to which, during aging, there would be a deficit in our body’s antioxidant defence system, and consequently cellular damage produced by free radicals. In older, prediabetic patients, lower activity of the enzyme Superoxide dismutase-1 and higher oxidative stress have been detected [84]. Oxidative stress has been linked to systemic inflammation, with both linked to insulin resistance [85]. Furthermore, as we age, changes occur in our body’s composition, with a loss of muscle tissue and an increase in fatty tissue [86]. These changes also influence insulin sensitivity, by reducing the entry of glucose into the muscles and producing oxidative stress and a pro-inflammatory response via the fatty tissue [87]. In the multivariate analysis of our study, we observed that when stratifying the population according to age, the OR increases for each decade that has passed, such that the risk of developing IR increases from 20 to 60 years of age from 50% to more than 100%, depending on the formula used.

Regarding the educational level, we found no differences in the insulin resistance risk scales in our population. In our bibliographic review, we did not find any studies that detect an association between participants’ educational level and IR. Stephens et al. [88], in their study on educational level and metabolic disorders, found a relationship between IR and educational level, but it lacked statistical significance. We found some studies that present an association between the educational level of the parents and the IR of their children [89], or the educational level of the parents and the metabolic syndrome [90]; however, our study consisted of an adult population and we did not assess the educational level of their parents. Some studies link primary or low levels of education and unhealthy behaviours, such as an increase in metabolic syndrome: people with a lower level of education had a higher risk of metabolic syndrome, and a lower probability of participating in health education and disease prevention activities [91].

Regarding social class, multiple studies have been published assessing socioeconomic levels and a greater increase in risk factors for non-communicable diseases in classes with a low socioeconomic level [92,93,94,95], which increases cardiometabolic risk and premature mortality; however, there are very few studies that relate IR to socioeconomic levels. Lopez-Jaramillo et al. published a study in 2023 in which they found that the population of countries with a medium and low socioeconomic level had a higher risk of developing IR [96]. These authors used the TyG index to identify IR, one of the indices that was used in our study. Our work also found a relationship between a medium–low socioeconomic level and an increase in IR; however, in our case, the population all came from the same country. We consider this difference to be important, since the previous study established the possibility that the differences were due to the ability to access the health systems of each country. In our study, this likelihood is nullified, as we are dealing with a Spanish population with access to the same universal health care system. Another study also established an association between IR and a low socioeconomic level, although its design sought to relate IR or DM2 with noise pollution [97].

All the variables analysed—sociodemographic and healthy habits—influence the average values and the prevalence of high values of all IR risk scales.

In the multivariate analysis, all variables, except educational level, increase the risk of presenting high values on the IR risk scales, especially a sedentary lifestyle and a low adherence to the Mediterranean diet.

Finally, our study finds an association between IR, lack of regular physical exercise, smoking, unhealthy eating (with abundant saturated fats and processed foods) that is cheaper, and a low socioeconomic status. The latter influences all the other variables, as described in 1999 by David Gordon [98,99] in his work which established a relationship between poverty and poor health. This has subsequently been ratified in other works by the same author [100], and by the Townsend Centre for International Poverty Research of Bristol University, which, given the magnitude and influence of this problem, defined the Ten Alternative Tips for Better Health as a critique of the lack of solutions at all levels to this situation. The current global economic situation, the increase in differences between social classes, and the different multicultural regions make a holistic approach that takes socioeconomic, cultural, socioecological and political factors into account essential, which are increasingly necessary to implement multidimensional programs and public health and integrated community interventions to effectively prevent T2DM.

## 5. Strengths and Limitations

The strengths of this study comprise the enormous size of the sample, which exceeds 386,000 workers—thereby providing great power to the results obtained—and the great variety of insulin resistance risk scales employed.

On the other hand, its main limitation is that insulin resistance was not determined using objective or analytical methods, but rather using risk scales.

Our study paves the way for future research between socioeconomic levels and IR, as well as the association between educational levels and IR, where we have not found differences. These studies could be carried out in smaller samples that allow the use of a direct measurement method of insulin resistance, such as the HOMA-IR (insulin resistance index).

## 6. Conclusions

Our study provides a sufficiently large sample to demonstrate an association between the practice of regular physical exercise and a reduction in the risk of IR, a strong role of the Mediterranean diet as a protective factor against IR, as well as an association between aging and increased IR, which has been suggested in other studies, and, finally, a relationship between a low socioeconomic level and an increase in IR.

The profile of a person at high risk of developing IR would consist of them having the following characteristics: male, elderly, from social class III, smoker, sedentary, with a low adherence to the Mediterranean diet, and having a low socioeconomic level.

Among other risk factors for IR, a low socioeconomic status can significantly increase the risk of prediabetes and T2DM but is often overlooked. In multinational and multicultural regions, such as Europe, a holistic approach that takes into account both traditional and socioeconomic/socioecological factors is becoming increasingly important for the implementation of multidimensional public health programs and integrated community interventions for the effective prevention of T2DM.

## Figures and Tables

**Figure 1 nutrients-15-05122-f001:**
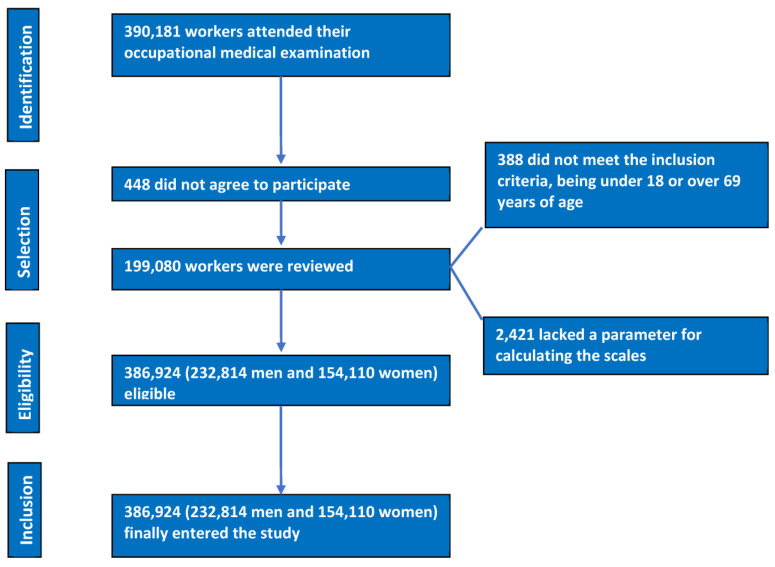
PRISMA flow chart of participants.

**Table 1 nutrients-15-05122-t001:** Characteristics of the population.

	Men *n* = 232,814	Women *n* = 154,110	
	Mean (SD)	Mean (SD)	*p*-Value
Age (years)	39.8 (10.3)	39.2 (10.2)	<0.001
Height (cm)	173.9 (7.0)	161.2 (6.6)	<0.001
Weight (kg)	81.1 (13.9)	65.3 (13.2)	<0.001
Waist circumference (cm)	87.7 (9.1)	73.9 (7.9)	<0.001
Hip circumference (cm)	100.0 (8.4)	97.2 (8.9)	<0.001
Systolic blood pressure (mmHg)	124.4 (15.1)	114.4 (14.8)	<0.001
Diastolic blood pressure (mmHg)	75.4 (10.6)	69.7 (10.3)	<0.001
Total cholesterol (mg/dL)	195.9 (38.9)	193.6 (36.4)	<0.001
HDL-c (mg/dL)	51.0 (7.0)	53.7 (7.6)	<0.001
LDL-c (mg/dL)	120.5 (37.6)	122.3 (37.0)	<0.001
Triglycerides (mg/dL)	123.8 (88.0)	88.1 (46.2)	<0.001
Glycaemia (mg/dL)	88.1 (12.9)	84.1 (11.5)	<0.001
	**%**	**%**	*p*-value
20–29 years	17.9	19.5	<0.001
30–39 years	33.1	33.3	
40–49 years	29.7	29.4	
50–59 years	16.3	15.3	
60–69 years	3.0	2.5	
Primary school	61.2	51.8	<0.001
Secondary school	34.0	40.7	
University	4.8	7.5	
Social class I	5.3	7.2	<0.001
Social class II	17.4	33.2	
Social class III	77.3	59.8	
Non-physical activity	54.5	47.8	<0.001
Yes, physical activity	45.5	52.2	
Non-healthy food	59.0	48.6	<0.001
Healthy food	41.0	51.4	
Non-smokers	62.9	67.0	<0.001
Smokers	37.1	33.0	

HDL-c, high-density lipoprotein cholesterol; LDL-c, low-density lipoprotein cholesterol; SD, Standard deviation.

**Table 2 nutrients-15-05122-t002:** Mean values of the insulin resistance risk scales according to sociodemographic variables and healthy habits by sex.

			Men					Women		
		TyG Index	TyG-BMI	METS-IR	TG/HDL-c		TyG Index	TyG-BMI	METS-IR	TG/HDL-c
	*n*	Mean (SD)	Mean (SD)	Mean (SD)	Mean (SD)	*n*	Mean (SD)	Mean (SD)	Mean (SD)	Mean (SD)
20–29 years	41,742	8.1 (0.5)	204.3 (40.3)	34.9 (6.8)	1.8 (1.4)	29,978	8.0 (0.5)	190.5 (42.3)	32.5 (7.2)	1.4 (0.8)
30–39 years	76,960	8.4 (0.6)	222.2 (42.0)	38.0 (7.1)	2.4 (2.1)	51,392	8.0 (0.5)	197.9 (45.1)	34.0 (7.7)	1.5 (0.9)
40–49 years	69,068	8.5 (0.6)	235.0 (43.1)	40.3 (7.4)	2.9 (2.5)	45,296	8.1 (0.5)	209.9 (45.6)	36.2 (7.7)	1.8 (1.0)
50–59 years	38,028	8.6 (0.6)	241.3 (41.8)	42.0 (7.3)	3.1 (2.4)	23,516	8.3 (0.5)	222.9 (46.0)	38.5 (7.7)	2.1 (1.3)
60–69 years	7016	8.6 (0.5)	245.0 (39.4)	42.9 (7.0)	3.1 (2.0)	3928	8.4 (0.5)	231.2 (43.7)	39.9 (7.3)	2.2 (1.1)
Primary school	142,494	8.4 (0.6)	226.8 (44.7)	39.0 (7.7)	2.6 (2.9)	79,810	8.1 (0.5)	211.3 (48.2)	36.4 (8.2)	1.7 (1.0)
Secondary school	79,226	8.4 (0.6)	226.4 (42.8)	38.8 (7.4)	2.5 (2.1)	62,690	8.1 (0.5)	198.4 (43.2)	34.0 (7.3)	1.6 (1.0)
University	11,094	8.3 (0.5)	224.0 (39.4)	38.5 (6.9)	2.5 (2.3)	11,610	8.0 (0.5)	193.0 (41.1)	33.1 (7.0)	1.6 (0.8)
Social class I	12,262	8.3 (0.5)	224.6 (40.2)	38.6 (7.0)	2.5 (2.2)	10,744	8.0 (0.5)	192.6 (40.5)	33.0 (6.9)	1.6 (0.8)
Social class II	40,650	8.4 (0.6)	225.5 (42.0)	38.6 (7.3)	2.5 (2.1)	51,230	8.1 (0.5)	195.5 (41.8)	33.6 (7.1)	1.6 (1.0)
Social class III	179,902	8.4 (0.6)	226.9 (44.4)	39.0 (7.7)	2.6 (2.2)	92,136	8.1 (0.5)	211.1 (48.1)	36.3 (8.2)	1.7 (1.0)
Non-physical activity	126,808	8.7 (0.6)	253.3 (39.5)	43.5 (6.9)	3.4 (2.7)	73,684	8.3 (0.5)	235.9 (46.3)	40.6 (7.8)	2.2 (1.2)
Yes, physical activity	106,006	8.1 (0.4)	194.5 (21.6)	33.4 (3.6)	1.6 (0.6)	80,426	7.9 (0.4)	176.0 (20.5)	30.2 (3.5)	1.3 (0.4)
Non-Mediterranean diet	137,464	8.7 (0.6)	249.0 (41.2)	42.7 (7.3)	3.3 (2.6)	74,828	8.3 (0.5)	233.2 (48.2)	40.0 (8.2)	2.1 (1.2)
Yes, Mediterranean diet	95,350	8.1 (0.4)	194.2 (21.7)	33.4 (3.6)	1.6 (0.6)	79,282	7.9 (0.4)	177.7 (21.7)	30.6 (3.7)	1.3 (0.5)
Non-smokers	146,480	8.4 (0.6)	228.7 (43.0)	39.2 (7.3)	2.4 (1.8)	103,300	8.1 (0.5)	207.2 (46.9)	35.6 (8.0)	1.7 (1.0)
Smokers	86,334	8.5 (0.6)	223.0 (44.9)	39.4 (8.0)	2.9 (2.7)	50,810	8.1 (0.5)	209.5 (44.5)	36.2 (7.5)	1.8 (1.1)

TyG index, triglyceride glucose index; BMI, body mass index; METS-IR, metabolic score for insulin resistance; TG, triglyceride; HDL-c, high-density lipoprotein cholesterol; SD, Standard deviation.

**Table 3 nutrients-15-05122-t003:** Prevalence of high values of insulin resistance risk scales according to sociodemographic variables and healthy habits by sex.

			Men					Women		
		TyG Index High	TyG-BMI High	METS-IR High	TG/HDL-c High		TyG Index High	TyG-BMI High	METS-IR High	TG/HDL-c High
	*n*	%	%	%	%	*n*	%	%	%	%
20–29 years	41,742	10.7	10.7	3.6	9.7	29,978	5.9	8.2	3.3	12.8
30–39 years	76,960	20.3	18.6	6.6	18.7	51,392	7.4	11.0	4.5	15.4
40–49 years	69,068	30.6	28.0	10.0	28.4	45,296	12.3	14.8	6.4	22.2
50–59 years	38,028	35.0	33.3	13.3	34.1	23,516	20.3	20.5	8.2	32.3
60–69 years	7016	36.6	36.7	14.5	34.1	3928	26.1	25.7	10.4	41.1
Primary school	142,494	25.2	23.5	8.9	23.5	79,810	12.7	16.7	6.9	22.2
Secondary school	79,226	23.8	22.2	7.8	22.2	62,690	9.5	10.2	4.2	18.2
University	11,094	20.9	19.9	6.6	20.6	11,610	7.8	7.7	3.2	16.4
Social class I	12,262	21.3	20.9	7.0	20.4	10,744	7.5	7.6	3.1	15.9
Social class II	40,650	23.7	21.4	7.3	22.4	51,230	9.3	9.0	3.7	18.3
Social class III	179,902	24.9	23.4	8.8	23.2	92,136	12.3	16.5	6.8	21.6
Non-physical activity	126,808	43.3	42.0	15.4	41.3	73,684	22.5	28.0	11.6	38.2
Yes, physical activity	106,006	2.0	0.3	2.4	0.9	80,426	0.4	0.3	0.2	3.5
Non-Mediterranean diet	137,464	39.8	38.8	14.2	37.8	74,828	21.2	27.6	11.4	34.5
Yes, Mediterranean diet	95,350	2.4	0.1	0.3	1.5	79,282	1.3	0.1	0.1	6.5
Non-smokers	146,480	22.6	23.8	8.2	20.9	103,300	10.6	14.6	6.1	19.4
Smokers	86,334	27.7	24.4	8.8	26.5	50,810	11.8	15.0	6.4	21.6

TyG index, triglyceride glucose index; BMI, body mass index; METS-IR, metabolic score for insulin resistance; TG, triglyceride; HDL-c, high-density lipoprotein cholesterol.

**Table 4 nutrients-15-05122-t004:** Multinomial logistic regression.

	METS-IR High	TG/HDL High	TyG Index High	TyG-BMI Index High
	OR (95% CI)	OR (95% CI)	OR (95% CI)	OR (95% CI)
Female	1	1	1	1
Male	1.25 (1.22–1.29)	1.01 (1.00–1.03)	2.57 (2.52–2.63)	1.67 (1.63–1.70)
20–29 years	1	1	1	1
30–39 years	1.11 (1.04–1.18)	1.11 (1.06–1.16)	1.14 (1.09–1.20)	1.14 (1.08–1.19)
40–49 years	1.21 (1.13–1.28)	1.30 (1.24–1.36)	1.28 (1.22–1.34)	1.17 (1.12–1.23)
50–59 years	1.33 (1.25–1.42)	1.64 (1.57–1.72)	1.66 (1.58–1.74)	1.31 (1.25–1.38)
60–69 years	1.54 (1.44–1.66)	2.08 (1.98–2.19)	2.22 (2.11–2.34)	1.48 (1.40–1.56)
Social class I	1	1	1	1
Social class II	1.19 (1.13–1.24)	1.04 (1.00–1.08)	1.08 (1.04–1.14)	1.20 (1.16–1.24)
Social class III	1.25 (1.17–1.33)	1.18 (1.16–1.20)	1.09 (1.06–1.11)	1.22 (1.16–1.27)
Yes, physical activity	1	1	1	1
Non-physical activity	21.10 (12.91–34.42)	31.32 (29.80–32.92)	22.58 (21.31–23.91)	78.77 (69.73–90.53)
Yes, Mediterranean diet	1	1	1	1
Non-Mediterranean diet	16.61 (8.32–33.34)	11.17 (10.80–13.58)	1.86 (1.77–1.96)	37.28 (22.55–54.88)
Non-smokers	1	1	1	1
Smokers	1.06 (1.03–1.09)	1.63 (1.60–1.66)	1.56 (1.53–1.59)	1.09 (1.07–1.11)

TyG index, triglyceride glucose index; BMI, body mass index; METS-IR, metabolic score for insulin resistance; TG, triglyceride; HDL-c, high-density lipoprotein cholesterol; OR, odds ratio; CI, confidence interval.

## Data Availability

This study’s data are stored in a database that complies with all security measures at the ADEMA-Escuela Universitaria. The Data Protection Delegate is Ángel Arturo López González.

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
