# Peer review of "Influence of Sociodemographic Variables and Healthy Habits on the Values of Insulin Resistance Indicators in 386,924 Spanish Workers"

_nutrients, 2023, doi:10.3390/nu15245122_

Round 1

Reviewer 1 Report

Comments and Suggestions for Authors

Dear Authors,

The article refers the important and current problem, and his medical implications cause  that it contains in the thematic profile of the periodical Nutrients. The work has an experimental character, carrying in relatively new cognitive elements from the sphere of basic sciences.

In general, the manuscript is well written and the experimental quality and the conclusions drawn are adequate.

However, there are the major concerns:

1/    The overall relevance of their paper is moderate. Several other groups have already analysed the prevalence of IR and suggested that the lack of physical activity (16 302 records in PubMed), age (36 259 records) and tobacco smoking may have a substantial effect on the development of  IR, with sometimes conflicting results.

2/ What was the purpose of measuring hip circumference if this parameter was not included in the analysis? If we had both waist and hip circumference, it was possible to use the WHR index, which, according to some researchers, correlates better with visceral obesity than waist circumference alone. Insulin resistance is associated with impaired secretion of substances by this adipose tissue that influence the sensitivity of peripheral tissues to insulin.

3/ Was a statistical test of agreement performed to check whether the groups were equal in size? Some statistical tests, including the Student's t-test used in the study, require that the groups compared are equal. However, there are a number of statistical tests that do not require such an assumption, e.g. the Kruskal-Wallis test, the Mann-Whitney U test, but this in turn implies the use of statistical measures other than mean and SD.

I have included all my comments in the attached manuscript.

Conclusion

Among other risk factors for IR, low socioeconomic status can significantly increase the risk of prediabetes and T2DM, but is often overlooked. In multinational and multicultural regions such as Europe, a holistic approach that takes into account both traditional and socioeconomic/socioecological factors is becoming increasingly important to implement multidimensional public health programs and integrated community interventions for the effective prevention of T2DM.

Author Response

Dear reviewer,

First of all, thank you for your work and all your recommendations.

To facilitate your review, we have written the modifications in red in the article.

The article refers the important and current problem, and his medical implications cause  that it contains in the thematic profile of the periodical Nutrients. The work has an experimental character, carrying in relatively new cognitive elements from the sphere of basic sciences.

In general, the manuscript is well written and the experimental quality and the conclusions drawn are adequate.

However, there are the major concerns:

1/    The overall relevance of their paper is moderate. Several other groups have already analysed the prevalence of IR and suggested that the lack of physical activity (16 302 records in PubMed), age (36 259 records) and tobacco smoking may have a substantial effect on the development of  IR, with sometimes conflicting results.

We completely agree with you that there are already multiple publications linking IR with physical activity, age and smoking. This part of our study aims to confirm these findings in a large sample, since they are part of the sociodemographic variables and healthy lifestyle habits. However, we have found very few works that study the association between educational level and socioeconomic level. Therefore, we believe that this is a novel and important contribution to our work. Which can open new avenues of research.

2/ What was the purpose of measuring hip circumference if this parameter was not included in the analysis? If we had both waist and hip circumference, it was possible to use the WHR index, which, according to some researchers, correlates better with visceral obesity than waist circumference alone. Insulin resistance is associated with impaired secretion of substances by this adipose tissue that influence the sensitivity of peripheral tissues to insulin.

The hip circumference measurement is included in the sample characteristics to give a more complete idea of the population.

The waist/hip ratio was not included, because another of our doctoral students is carrying out his doctoral thesis with different indicators related to the hip, such as the body adiposity index, the abdominal volume index and the waist/hip index.

3/ Was a statistical test of agreement performed to check whether the groups were equal in size? Some statistical tests, including the Student's t-test used in the study, require that the groups compared are equal. However, there are a number of statistical tests that do not require such an assumption, e.g. the Kruskal-Wallis test, the Mann-Whitney U test, but this in turn implies the use of statistical measures other than mean and SD.

Kolmogorov-Smirnov normality tests have been performed on the variables and it is observed that they have a normal distribution, for this reason the t-student test was applied.

I have included all my comments in the attached manuscript.

Conclusion

Among other risk factors for IR, low socioeconomic status can significantly increase the risk of prediabetes and T2DM, but is often overlooked. In multinational and multicultural regions such as Europe, a holistic approach that takes into account both traditional and socioeconomic/socioecological factors is becoming increasingly important to implement multidimensional public health programs and integrated community interventions for the effective prevention of T2DM.

The conclusion has been written according to your suggestions, many thanks.

Thank you very much for your suggestions. We have proceeded to answer all of them and we trust that they will adequately respond to your questions.

Reviewer 2 Report

Comments and Suggestions for Authors

The abstract gives a clear overview of the study, highlighting the sample size, participant demographics, and the utilization of multiple insulin resistance (IR) risk scales. Mentioning the limitations, particularly the lack of objective measures for insulin resistance, is crucial for transparency.

Introduction: The introduction effectively outlines the significance of studying insulin resistance and establishes a connection to lifestyle factors. The rationale for using various IR risk scales is justified, considering the impracticality of employing the HOMA-IR method due to the extensive sample size.

Methods: The study's strength lies in its massive sample size, providing substantial statistical power. Stratifying participants by age, education, and social class enhances the study's comprehensiveness. The utilization of validated IR risk scales is justified. However, the absence of direct measurements for insulin resistance is acknowledged as a limitation.

Results: The results section is well-organized, presenting a detailed analysis of the data. The authors effectively discuss the impact of sociodemographic variables and lifestyle choices on insulin resistance risk. The use of multinomial logistic regression adds depth to the analysis.

Discussion: The discussion section delves into the implications of the findings. The study's alignment with existing literature on smoking, physical activity, and dietary habits influencing insulin resistance strengthens its contribution. The discussion on socioeconomic factors and their association with insulin resistance adds a novel dimension.

Conclusion: The conclusion succinctly summarizes the key findings, emphasizing the association between a sedentary lifestyle, poor adherence to the Mediterranean diet, and increased insulin resistance risk. The limitations are appropriately reiterated.

Strengths and Limitations: The strengths of the study, such as the extensive sample size and the variety of insulin resistance risk scales used, are duly highlighted. The main limitation, the lack of direct insulin resistance measurements, is acknowledged, emphasizing the need for future research with objective methods.

Overall Impression: The paper contributes valuable insights into insulin resistance in a Spanish population. The comprehensive analysis of various risk factors provides a holistic understanding. The study's strengths, particularly the large sample size, enhance the robustness of the findings. The acknowledgment of limitations adds transparency to the research.

Recommendations: Consider expanding the discussion on the potential implications of the findings for public health interventions. Additionally, addressing the limitations by proposing avenues for future research could strengthen the paper further.

Please note that this is a general overview, and a more detailed review would require a thorough examination of each section of the paper.

Recheck the figures: some are unshaped, such as figure 1

Comments on the Quality of English Language

moderate editing of English language required

Author Response

Dear reviewer,

First of all, thank you for your work and all your recommendations.

To facilitate your review, we have written the modifications in red in the article.

The abstract gives a clear overview of the study, highlighting the sample size, participant demographics, and the utilization of multiple insulin resistance (IR) risk scales. Mentioning the limitations, particularly the lack of objective measures for insulin resistance, is crucial for transparency.

Introduction: The introduction effectively outlines the significance of studying insulin resistance and establishes a connection to lifestyle factors. The rationale for using various IR risk scales is justified, considering the impracticality of employing the HOMA-IR method due to the extensive sample size.

Methods: The study's strength lies in its massive sample size, providing substantial statistical power. Stratifying participants by age, education, and social class enhances the study's comprehensiveness. The utilization of validated IR risk scales is justified. However, the absence of direct measurements for insulin resistance is acknowledged as a limitation.

Results: The results section is well-organized, presenting a detailed analysis of the data. The authors effectively discuss the impact of sociodemographic variables and lifestyle choices on insulin resistance risk. The use of multinomial logistic regression adds depth to the analysis.

Discussion: The discussion section delves into the implications of the findings. The study's alignment with existing literature on smoking, physical activity, and dietary habits influencing insulin resistance strengthens its contribution. The discussion on socioeconomic factors and their association with insulin resistance adds a novel dimension.

Conclusion: The conclusion succinctly summarizes the key findings, emphasizing the association between a sedentary lifestyle, poor adherence to the Mediterranean diet, and increased insulin resistance risk. The limitations are appropriately reiterated.

Strengths and Limitations: The strengths of the study, such as the extensive sample size and the variety of insulin resistance risk scales used, are duly highlighted. The main limitation, the lack of direct insulin resistance measurements, is acknowledged, emphasizing the need for future research with objective methods.

Overall Impression: The paper contributes valuable insights into insulin resistance in a Spanish population. The comprehensive analysis of various risk factors provides a holistic understanding. The study's strengths, particularly the large sample size, enhance the robustness of the findings. The acknowledgment of limitations adds transparency to the research.

Recommendations: Consider expanding the discussion on the potential implications of the findings for public health interventions. Additionally, addressing the limitations by proposing avenues for future research could strengthen the paper further.

Following their recommendations, we have proceeded to expand the discussion by commenting on the implications of the results on public health, reflections to take into account and actions to be implemented. Likewise, we have suggested new avenues of research as continuity to our findings.

Thank you very much.

Please note that this is a general overview, and a more detailed review would require a thorough examination of each section of the paper.

Recheck the figures: some are unshaped, such as figure 1

Thank you so much. We have proceeded to reconfigure the figure.

Thank you very much for your suggestions. We have proceeded to answer all of them and we trust that they will adequately respond to your questions.

Round 2

Reviewer 2 Report

Comments and Suggestions for Authors

no other suggestions

Comments on the Quality of English Language

Minor editing of English language required